# Thalidomide Alleviates Pulmonary Fibrosis Induced by Silica in Mice by Inhibiting ER Stress and the TLR4-NF-κB Pathway

**DOI:** 10.3390/ijms23105656

**Published:** 2022-05-18

**Authors:** Yaqian Li, Wenchen Cai, Fuyu Jin, Xiaojing Wang, Wenjing Liu, Tian Li, Xinyu Yang, Heliang Liu, Hong Xu, Fang Yang

**Affiliations:** Hebei Key Laboratory for Organ Fibrosis Research, School of Public Health, North China University of Science and Technology, Tangshan 063210, China; lyqewbar@163.com (Y.L.); chenwencai1991@163.com (W.C.); fuyujinjfy@163.com (F.J.); wxj15031902829wxj@163.com (X.W.); liuwenjing_1997@163.com (W.L.); tiantian__1997@163.com (T.L.); zwns69618@163.com (X.Y.); 13933499300@163.com (H.L.)

**Keywords:** thalidomide, ER stress, TLR4, inflammation, macrophages, silicosis

## Abstract

Silicosis is the most prevalent occupational disease in China. It is a form of pulmonary fibrosis caused by the inhalation of silicon particles. As there is no cure for the potentially lethal and progressive condition, the treatment of silicotic fibrosis is an important and difficult problem to address. Thalidomide, a drug with anti-inflammatory and immunoregulatory properties, has been reported to have lung-protective effects. The purpose of this study was to observe the therapeutic effect of thalidomide on silicotic mice and to determine the protective mechanism. By using silicotic mice models and MH-S cells, we found the expression of endoplasmic reticulum stress (ER stress) and Toll-like receptor 4 (TLR4)-nuclear factor kappa-B (NF-κB) pathway as well as inflammation-related factors were upregulated in the macrophages of silicotic mice. The same indexes were detected in silica-stimulated MH-S cells, and the results were consistent with those in vivo. That is, silica activated ER stress and the TLR4-NF-κB pathway as well as the inflammatory response in vitro. Treating both silicotic mice and silica-stimulated MH-S cells with thalidomide inhibited ER stress and the TLR4-NF-κB pathway as well as the inflammatory response. The present study demonstrates thalidomide as a potential therapeutic agent against silicosis.

## 1. Introduction

Silicosis, which is caused by inhalation of silica, is the most prevalent occupational disease in China, leading to progressive, irreversible, and fatal inflammation and fibrosis of the lung [1]. The cause of silicosis is well known, yet there is currently no effective treatment for the disease. Many drug targets have proven to be effective in in vitro studies, and animal experiments present the next logical step in testing these candidates [2,3]. Some drugs have been approved to treat pulmonary fibrotic disease, including pirfenidone and nintedanib [4] in the United States and tetrandrine [5] in China. However, these drugs have limited efficacy in preventing disease progression and improving patient quality of life, as well as being associated with tolerability issues [6,7]. Until now, lung transplantation has been the only way to cure pulmonary fibrosis [8]. The pathophysiological mechanism underpinning the occurrence and development of silicosis in the lungs remains unclear.

The thalidomide molecule is a racemic glutamic acid analogue, consisting of S(−) and R(+) enantiomers that interconvert under physiological conditions [9]. It has previously been prescribed as a sedative, tranquilizer, and antiemetic for morning sickness [10]. However, due to teratogenic effects on the fetus (phocomelia), thalidomide was once banned clinically [11], but later studies have since shown the compound to possess anti-inflammatory, immunomodulatory, and antiangiogenic properties, and hence to be effective in treating inflammatory conditions, infectious conditions, and malignant diseases [12]. Because of this, its ban was lifted in 1997 in the United States [13]. Lately, thalidomide has been reported to be useful in the treatment of pulmonary fibrosis, including idiopathic pulmonary fibrosis and radiation-induced lung fibrosis [14,15]. Despite these studies, the mechanisms involved remain to be fully investigated.

Research has shown that endoplasmic reticulum stress (ER stress) facilitates fibrotic remodeling through the activation of pro-apoptotic pathways, the induction of epithelial-mesenchymal transition, and the promotion of inflammatory responses [16]. Our previous studies discovered that both inflammation and ER stress play vital roles in the progression of silicosis [17,18,19,20]. It is worth noting that early studies on ER stress have focused on epithelial cells [16,18,19] with researchers since turning their attention to macrophages [17,20]. Previous data indicated that silica inhalation might activate the Toll-like receptor 4 (TLR4)-nuclear factor kappa-B (NF-κB) signaling pathway in lung macrophages [20]. Thalidomide inhibits lipopolysaccharide-induced tumor necrosis factor-a (TNF-α)-production via down-regulating the expression of the TLR4-downstream myeloid differentiation factor 88 (MyD88) [21]. Therefore, in the present study, the effects of thalidomide on silica-induced pulmonary fibrosis were investigated in mice. Here, we show that thalidomide prevented silica-caused pulmonary fibrosis in mice. The underlying regulatory mechanisms associated with the potential anti-inflammatory and anti-ER stress effects of thalidomide were also investigated.

## 2. Results

### 2.1. Thalidomide Reduced Collagen Deposition in Silicotic Mice

To establish the silicosis mouse model, we applied a single intratracheal instillation of a silica suspension in 8-week-old male C57BL/6 mice. Mice were then given intraperitoneal injections of thalidomide every day for 4 weeks until sacrifice (Figure 1A); we observed that treatment with the thalidomide overcame the suppression of lung functions in response to silica exposure (Figure 1B–F). Four weeks after tracheal perfusion, significant collagen deposition was observed in the sections of mice from the silica perfusion group; however, the pulmonary collagen deposition of mice was significantly inhibited after different concentrations of thalidomide were administered (Figure 2A–G). As a well-known antiangiogenic agent [22], thalidomide significantly reduced vascular endothelial growth factor (VEGF) expression in mice lung tissue (Figure 2D,H).

### 2.2. Thalidomide Inhibited ER Stress in Silicotic Mice

Previous studies have shown that ER stress was activated in rats affected by idiopathic pulmonary fibrosis and silicosis [19,23]. The expression and localization of ER stress-related proteins in silicotic mice were examined in the present study. In our silicotic mice model, the expression of phospho-PKR-like ER kinase (p-PERK), phospho-inositol-requiring enzyme 1α (p-IRE-1α), and phospho-eukaryotic initiation factor 2 alpha (p-eIF-2α) was significantly upregulated (Figure 3B), and we observed the macrophage marker CD68 [24] to be co-expressed with p-PERK (Figure 3A). This phenomenon showed that the activation of ER stress happened in macrophages. We found that both tested concentrations of thalidomide could effectively inhibit the activation of ER stress, and there appeared to be no difference between the two concentrations (Figure 3B–E).

### 2.3. Thalidomide Inhibited the TLR4-NF-κB Pathway and Inflammatory Response in Silicotic Mice

Separate from the silica inhalation model [25], we hypothesized that a single infusion comprised of a large volume of a silica suspension into the lungs would induce a strong inflammatory response with the associated signal activation, as mentioned in a previous report [26] and that thalidomide would reverse this activation. To test our hypothesis, we detected the activity of the TLR4-NF-κB pathway and measured inflammatory cytokines in the lungs of mice following 4 weeks of silica administration. As shown in Figure 4B,H, the levels of TLR4, MyD88, phospho-nuclear factor kappa-B (p-NF-κB), inhibitor of NF-κB α (p-IκBα), TNF-α, interleukin-6 (IL-6), and interleukin-1β (IL-1β) were significantly upregulated in mice treated with SiO_2_ and compared with the SiO_2_-infused group, thalidomide significantly inhibited the expression of these factors (Figure 4B–F,H–K). Results reflected those from the Western blot and tissue immunofluorescence staining; TLR4 and TNF-α were strongly expressed in the silicosis model group, and with the treatment of thalidomide, the expression levels of TLR4 and TNF-α were decreased (Figure 4A,G). It is worth noting that we observed CD68 to be co-expressed with TLR4 and TNF-α (Figure 4A,G). This phenomenon showed that the activation of the TLR4-NF-κB pathway and inflammatory-related factors occurred in macrophages.

### 2.4. SiO_2_ Stimulated the Activation of ER Stress and TLR4-NF-κB as Well as Inflammation-Related Factors in MH-S Cells

The in vivo results using mouse macrophages clearly indicated the activation of ER stress, the inflammatory response, and upregulation of TLR4-NF-κB. Therefore, we chose MH-S cells for subsequent studies. To explore the optimal concentration of SiO_2_ suspension to use on cells, we selected different concentrations of the SiO_2_ suspension (50, 100, and 200 μg/mL, respectively) to treat MH-S cells, and the expression of relevant proteins was then measured. The results of the Western blot were consistent with our expectations, namely that the expression of TLR4, MyD88, p-NF-ΚB, p-IκBα, TNF-α, IL-6, and IL-1β was increased after the cells were treated with different concentrations of the SiO_2_ suspension (Figure 5E–M). After treatment with 50 μg/mL SiO_2_ suspension, MH-S cells showed no statistically significant difference in levels of the p-IRE-1α protein between the control group and SiO_2_ group (Figure 5A,D). MH-S cells stimulated with other concentrations were statistically significant on measures of ER stress, TLR4-NF-κB expression, and levels of inflammation-related factors compared with the control group; however, the difference in protein expression between the three concentrations of SiO_2_ was not statistically significant (Figure 5B–D,F–I,K–M). Therefore, 100 μg/mL was selected as the concentration of silica used for subsequent experiments.

### 2.5. Thalidomide Inhibited ER Stress and the TLR4-NF-κB Pathway as Well as the Inflammatory Response in MH-S Cells Treated with SiO_2_

We found that treatment with SiO_2_ resulted in the upregulation of p-PERK, TLR4, and TNF-α (Figure 6A,B and Figure 7A,B,H,I). These images in combination with the results of Western blot (Figure 6C–F and Figure 7C–G,J–M) further validated our reasoning: ER stress, the TLR4-NF-κB pathway, and the inflammatory response were activated in macrophages. Then, we used thalidomide to treat MH-S cells. The results from immunofluorescence and Western blot showed that thalidomide significantly inhibited ER stress, inflammatory response activation, and the TLR4-NF-κB pathway in macrophages (Figure 6C and Figure 7C,J).

## 3. Discussion

This study was undertaken to explore the therapeutic effect of thalidomide on silicosis in mice and the potential underlying mechanisms. Here, our results demonstrated that treatment with thalidomide could reduce collagen deposition in silicotic mice. Specifically, we observed that the physiological response to silica exposure in the lungs involves upregulating ER stress, the inflammatory response, and the TLR4-NF-κB pathway and thalidomide could reverse this outcome. Thalidomide was required to maintain ECM homeostasis by inhibiting ER stress, the inflammatory response, and the TLR4-NF-κB pathway response in vitro. Collectively, our work lends strong evidence to support the benefits of thalidomide in limiting ER stress, the inflammatory response, and the TLR4-NF-κB pathway activation accompanying silicosis and provides valuable insights into the mechanisms underlying the attenuation of this disease.

In elucidating how thalidomide can inhibit collagen deposition, we focused on previous reports. Accumulating evidence has indicated that ER stress is a possible molecular mechanism causing fibrosis in a variety of organs [27,28,29]. In particular, ER stress seems to play an important role in silicosis fibrosis [17,19,30]. The ER plays a major role in orchestrating biogenesis through the synthesis, folding, assembly, trafficking, and structural maturation of nearly a third of all proteins made in the cell, as well as the degradation of defective proteins [31]. Errors in any one of these processes can trigger ER stress. As an inhaled toxic substance, SiO_2_ can cause DNA damage [32] which may also trigger ER stress. Accordingly, in the present study, silica increased the levels of p-PERK, p-eIF2α, and p-IRE1α both in silicotic mice and MH-S cells. Subsequently, treatment with thalidomide reversed the activation of ER stress caused by SiO_2_ both in vivo and in vitro. Despite being known as a human teratogen, thalidomide exhibits diverse therapeutic biological activity, being anti-inflammatory, immunomodulatory, and anti-angiogenic [12]. Nevertheless, there are few reports about the relationship between thalidomide and ER stress; such a connection was only made in one paper in 2021, reporting that thalidomide inhibited ER stress in type 2 diabetic mice and in MCs cells induced by high glucose [33], which appears to be supported by our present study on ER stress. Thalidomide inhibited collagen deposition by inhibiting ER stress-induced by silica. Conversely, lipid droplet-binding members of thalidomide analogs could induce ER stress by affecting ER-membrane integrity and perturbing ER homeostasis in cancer cells [34]. These findings suggest that the different roles of thalidomide may be related to different cell types and disease types.

As the most extensively studied of the mammalian family of Toll-like receptors, TLR4 can be activated by both LPS in infectious conditions and endogenous compounds called damage-associated molecular patterns (DAMPs) in non-infectious conditions to induce tissue repair [35]. TLR4 can recognize and bind with extracellular High Mobility Group Box 1 and activate the MyD88 pathway and NF-κB signaling pathway, thereby triggering a cascade effect. Accordingly, excessive pro-inflammatory cytokines and chemokines are produced to induce inflammation [36]. In the present study, treatment with SiO_2_ activated the TLR4-NF-κB pathway and triggered the inflammatory response. These results suggest that persistent inflammation in the lungs triggered by silica [20,37] can result in the necrosis and apoptosis of macrophages containing SiO_2_ and cause the release of intracellular components (via lysosome rupture) that subsequently act as DAMPs. Ultimately, the TLR4-NF-κB pathway was activated in macrophages. Consistent with a previous study [21], our results indicated that thalidomide reduced the level of MyD88, NF-κB, and TNF-α protein levels. We subsequently observed reductions in IL-6 and IL-1β levels. However, contrary to our results, an earlier study indicated that thalidomide promoted the release of TNF-α in mice [38]. This may be due to differences in mouse species or disease models. There is one similar case reported in 2009 which indicated that thalidomide did not affect the cell surface expression of TLR4 [21], a conclusion that was not reached in our study. As immunomodulatory drugs, thalidomide and its derivatives are effective inhibitors of TLR4-induced TNF-α production [39]. However, the previous literature did not elucidate whether thalidomide inhibits TLR4 expression. Nevertheless, our study suggests that the impact of thalidomide on TLR4 activity broadens the spectrum of pathophysiological conditions far beyond immunological diseases. Furthermore, angiotensin II (Ang II) has been reported to activate TLR4 as well as its downstream signaling cascade [40,41], and during dermatological follow-up with thalidomide, blood levels of the angiotensin-converting enzyme (ACE) decreased [42]. It is well known that a drop in ACE level causes a drop in Ang II level. The renin–angiotensin–aldosterone system also plays an important role in the progression of silicosis fibrosis [43]; thus, the above research results support the results of this study, namely, treatment with thalidomide can inhibit the expression of TLR4. TLR4 is also crucial to the development of ER stress [44,45]. Whether these two pathways have synergistic effects on the development of silicosis remains to be studied.

Widely used in the treatment of dermatological diseases, immune disorders, and cancers [12], thalidomide has also been reported to have lung-protective effects. In addition to the focus of this article on pulmonary fibrosis, it also plays a protective role in paraquat poisoning and lung damage caused by cigarette smoke [46,47]. Our study further broadens the scope of the protective effect of thalidomide on the lung, confirming its protective effect on silicosis. Many studies on the administration of thalidomide have adopted the method of intragastric administration or oral administration [15,33]. To improve the bioavailability of thalidomide, intraperitoneal injection was chosen in the present study. According to our results, both concentrations of thalidomide used appear to inhibit the progression of silicosis to the same extent (the difference between them was not statistically significant). Considering the possible side effects and efficacy of the drug, we would choose the lower dose in the follow-up study. However, due to its teratogenic effects, the use of thalidomide in female patients is highly restricted. To solve this problem, studies on the teratogenicity of thalidomide are also underway. By using cereblon-deficient mice, researchers have been able to demonstrate that certain immunological properties of thalidomide and its derivatives occur independently of cereblon, the mediator of thalidomide teratogenicity [48]. This discovery provides a new avenue for the development of thalidomide analogues.

## 4. Materials and Method

### 4.1. Establishment of the Mouse Model

Four-week-old male C57BL/6 mice were purchased from Vital River Laboratory Animal Technology. The mice were maintained in a light/dark cycle for 12 h and provided with free access to food and water. The study protocol was approved by the Committee on the Ethics of North China University of Science and Technology (LX2019033) and complied with the US National Institutes of Health Guide for the Care and Use of Laboratory Animals.

The mice were randomly divided into four groups (*n* = 8 each) as follows: (1) control group: subjected to tracheal perfusion with 50 μL 0.9% normal saline, (2) silica group: subjected to tracheal perfusion with 10 mg silica suspension (50 μL) [49] (s5631; Sigma–Aldrich, St. Louis, MO, USA), (3) silica plus thalidomide group: subjected to tracheal perfusion with 10 mg/ 50 μL silica suspension, then the following day with 100 μL 50 mg/kg thalidomide (H32028128, Changzhou Pharmaceutical Factory Co. LTD, Changzhou, China) was injected intraperitoneally, with continuous injections for 28 days, and (4) silica plus thalidomide (100 mg/kg) group: subjected to tracheal perfusion with 10 mg/ 50 μL silica suspension, then the following day with 100 μL 50 mg/kg thalidomide was injected intraperitoneally, with continuous injections for 28 days. Mice were then sacrificed, and part of the lung tissue was dehydrated and embedded in paraffin, while the remaining lung tissue was stored at −80 °C.

### 4.2. Cell Culture

MH-S cells were obtained from the Chinese Academy of Sciences Cell Bank (Shanghai, China) and cultured in RPMI 1640 medium (10-040-CV; Madiatech, lnc., Manassas, VA, USA) in Petri dishes placed in a 37 °C incubator containing 5% CO_2_. Cells were then given one of the following treatments with or without the conjunction with 100 μg/mL thalidomide: 50 μg/mL SiO_2_, 100 μg/mL SiO_2_, or 200 μg/mL SiO_2_.

### 4.3. Hematoxylin–Eosin Staining

Paraffin-embedded lung tissue sections were deparaffinized and rehydrated. Hematoxylin and eosin dye (BA4025, Baso Diagnostics Inc., Zhuhai, China) was then added dropwise to observe the histopathological morphology.

### 4.4. Van Gieson’s Staining

The rehydrated lung tissue sections were covered with an equal proportion of hematoxylin A and hematoxylin B compared to each other and then added with V.G. dye (BA4084, BaSO Diagnostics Inc., Zhuhai, China). The area of collagen stained by the V.G. dye was counted by Image-Pro Plus 6.0 software package (Media Cybernetics, Inc., Rockville, MD, USA) and then homogenized by the total area.

### 4.5. Western Blot

Western blotting was performed as described previously [3]. Primary antibodies included CoL I (ab34710; Abcam, Cambridge, UK), α SMA (ab5694; Abcam, Cambridge, UK), PAI-1 (BS3503; Bioword, Minnesota, Mn, USA), VEGF (ET1604-28; HuaAn, Hangzhou, China), p-PERK (DF7576; Affinity Biosciences, Cincinnati, OH, USA), PKR-like ER kinase (PERK) (ER64553; HuaAn, Hangzhou, China), p-IRE-1α(S724) (ab48187; Abcam, Cambridge, UK), inositol-requiring enzyme 1α (IRE-1α) (A00683-1; Boster, Wuhan, China), p-eIF-2α(S51) (ET1603-14; HuaAn, Hangzhou, China), eukaryotic initiation factor 2 alpha (eIF-2α) (ET7111-34; HuaAn, Hangzhou, China), TLR4 (ARG20515; arigo, Shanghai, China), MyD88 (ab2068; Abcam, Cambridge, UK), p-IκBα(S32/S36) (AF2002; Affinity Biosciences, Cincinnati, OH, USA), p-NF-κB (ARG51516; arigo, Shanghai, China), NF-κB (ARG51013; arigo, Shanghai, China), TNF-α (, GTX110520; GeneTex, San Antonio, TX, USA), IL-1β (DF6251;Affinity Biosciences, Cincinnati, OH, USA ), IL-6 (A0286; ABclonal, Wuhan, China), and β-actin (AC026; ABclonal, Wuhan, China). All antibodies were diluted at 1:1000. Incubation was then performed using goat anti-rabbit or anti-mouse secondary antibodies (074-1506/074-1806; Kirkegaard and Perry Laboratories, Gaithersburg, MD, USA) at a dilution of 1:5000 in blocking buffer. Immunoblot target bands were visualized using ECL prime Western blotting detection reagent (ZD310A; ZomanBio, Beijing, China). The results were normalized against the corresponding control.

### 4.6. Immunofluorescence Staining (IF)

Immunofluorescence staining was performed as previously described [20]. After the tissues and cells were fully hydrated, the antigen was exposed to high-pressure retrieval. The tissues and cells were respectively incubated with p-PERK/CD68 (ab201340, Abcam, Cambridge, UK), TLR4/CD68, and TNF-α/CD68 at a dilution of 1:200 overnight at 4 °C. Tissues and cells were then respectively combined with the secondary antibody at 37 °C for 40 min. The nuclei were stained with DAPI (8961s; Cell Signaling Technology, Inc., Danvers, MA, USA).

### 4.7. Lung Function Assessment

Respiratory parameters were assessed in whole body plethysmograph (WBP) chambers (FinePointe WBP, BUXCO Research Systems, INC, Wilmington, NC, USA), following the manufacturer’s protocol. Mice were placed into the whole body plethysmographic chambers. After a few minutes for stabilization, lung function detection included an adaptation period (10 min), atomization period (1 s), reaction period (5 min), and recovery period (1 min).

### 4.8. Statistical Analysis

Statistical analyses were performed using SPSS 20.0 software (IBM Corp., Armonk, NY, USA). The data were expressed as the mean ± standard deviation (SD). Multiple comparisons were performed using a one-way analysis of variance (ANOVA) followed by Tukey’s post-hoc test. Statistical significance was achieved when *p* < 0.05 at a 95% confidence interval.

## 5. Conclusions

Using an early administration strategy, our work demonstrated that thalidomide can effectively attenuate silica-induced pulmonary inflammation and fibrosis in mice. In addition, we report that thalidomide mediated the alleviation of silicosis via the inhibition of ER stress, the inflammatory response, and the TLR4-NF-κB (Figure 8). The results of our study provide a substantial conceptual framework for understanding the basic mechanisms by which thalidomide ameliorates silicosis. Furthermore, our work also provides strong evidence for clinical strategies that combine thalidomide with other inhibitors of ER stress, the inflammatory response, and the TLR4-NF-κB pathway to increase the efficacy and therapeutic range available for silicosis patients.

## Figures and Tables

**Figure 1 ijms-23-05656-f001:**
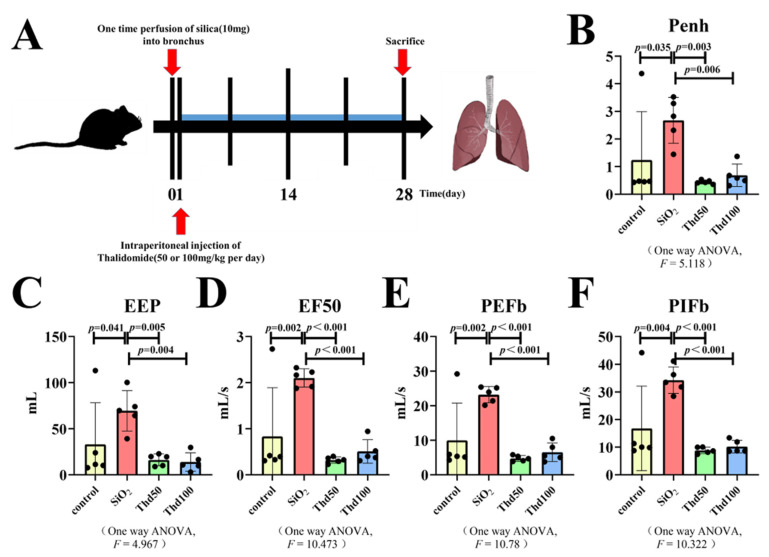
Treatment with the thalidomide overcame the suppression of lung functions in response to silica exposure. (**A**) Schematic of thalidomide administration in therapeutic administration model of silicosis mice. There are four groups (Control, SiO_2_, SiO_2_ + thalidomide 50, SiO_2_ + thalidomide 100) in the following experiments (8 mice per group). (**B**–**F**) Lung functions of mice exposed to silica and treated with thalidomide. Data are presented as the mean ± SD. *n* = 5 per group.

**Figure 2 ijms-23-05656-f002:**
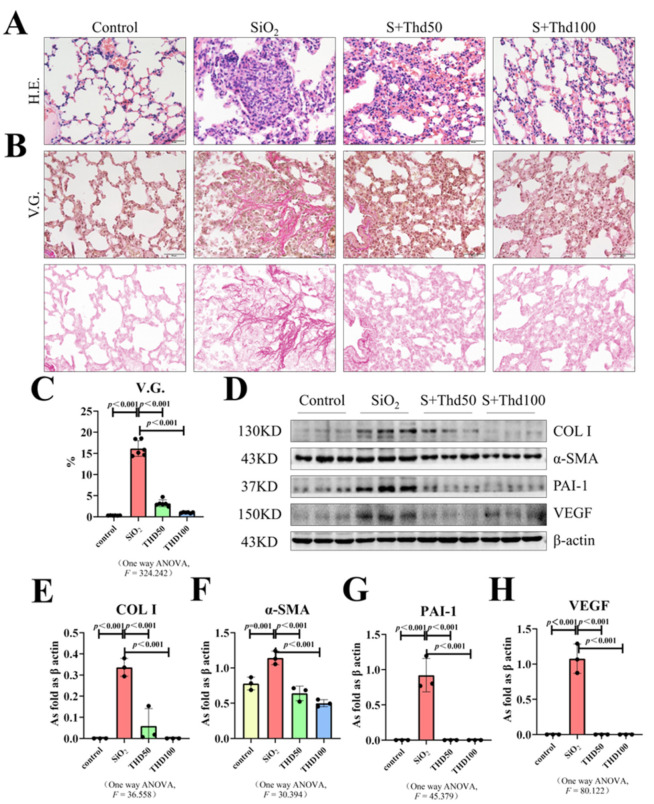
Thalidomide inhibited the progression of fibrosis in silicosis mice. (**A**) HE staining of lung tissue in mice exposed to silica (scale bar = 50 µm). (**B**,**C**) V.G. staining of lung tissue in mice exposed to silica (scale bar = 50 µm). (**D**–**H**) Expression levels of CoL I, α-smooth muscle actin (α-SMA), plasminogen activator inhibitor type-1 (PAI-1), and VEGF in the lungs of mice measured by Western blot. Data are presented as the mean ± SD, *n* = 3 per group.

**Figure 3 ijms-23-05656-f003:**
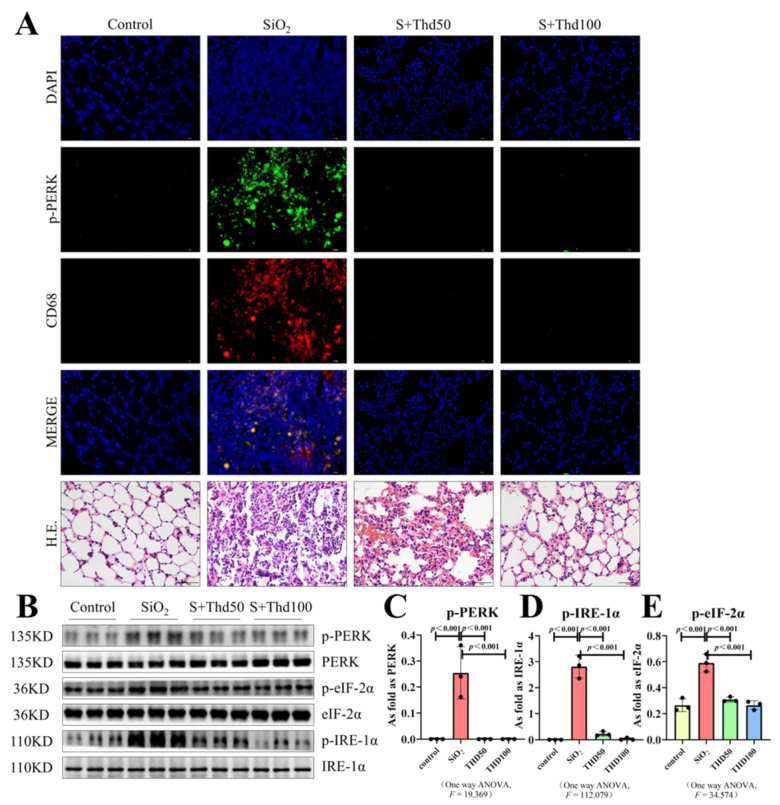
Thalidomide inhibited the ER stress signal in silicosis mice. (**A**) Expression of p-PERK in silicosis mice measured by IF staining (scale bar = 50 µm). (**B**–**E**) Expression levels of p-PERK, p-eIF-2α, and p-IRE-1α in the lungs of mice measured by Western blot. Data are presented as the mean ± SD, *n* = 3 per group.

**Figure 4 ijms-23-05656-f004:**
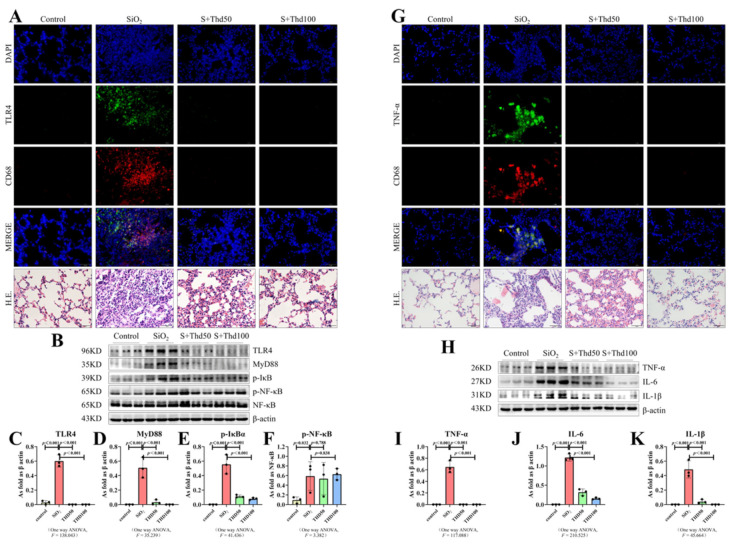
Thalidomide inhibited the expression of the TLR4 signal and inflammatory factors in silicosis mice. (**A**,**G**) Expression of TLR4 and TNF-α in silicosis mice measured by IF staining (scale bar = 50 µm). (**B**–**F**,**H**–**K**) Expression levels of TLR4, MyD88, p-IκBα, p-NF-κB, TNF-α, IL-6, and IL-1β in the lungs of mice measured by Western blot. Data are presented as the mean ± SD, *n* = 3 per group.

**Figure 5 ijms-23-05656-f005:**
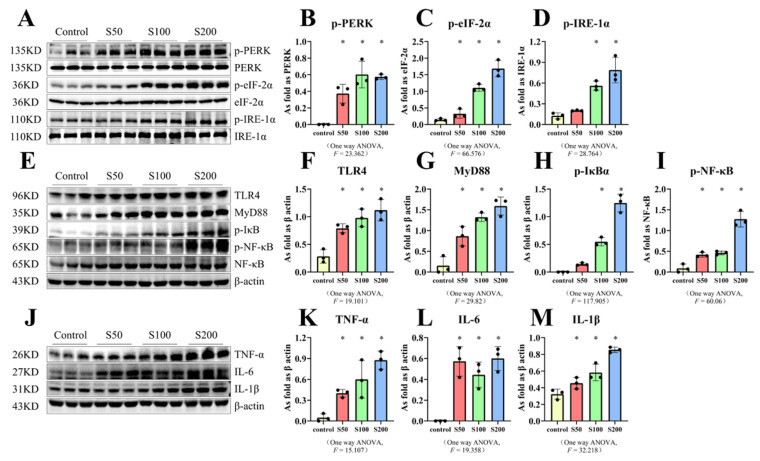
Silica activated the ER stress, TLR4-NF-κB pathway as well as inflammatory response in macrophages. (**A**–**M**) Expression levels of p-PERK, p-eIF-2α, p-IRE-1α, TLR4, MyD88, p-IκBα, p-NF-κB, TNF-α, IL-6, and IL-1β in MH-S cells treated with SiO_2_ at different doses measured by Western blot. * Compared with control group, *p* < 0.05. Data are presented as the mean ± SD, *n* = 3 per group.

**Figure 6 ijms-23-05656-f006:**
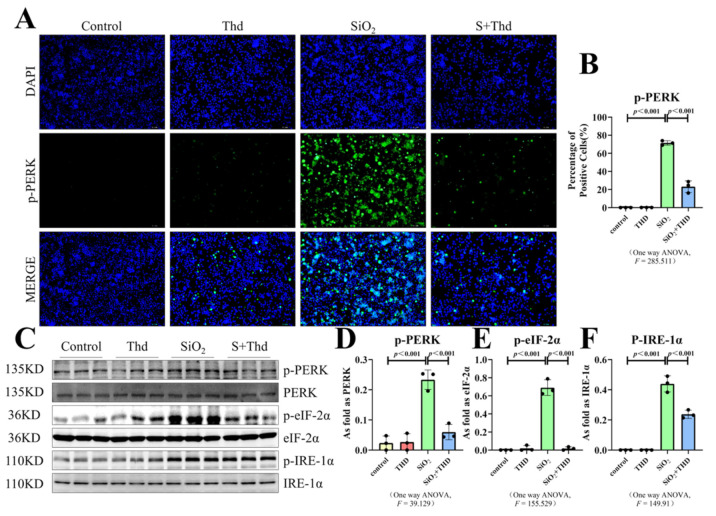
Thalidomide inhibited the ER stress signal in macrophages. (**A**,**B**) Expression of p-PERK in MH-S cells observed by IF staining (scale bar = 100 µm). (**C**–**F**) Protein expression of p-PERK, p-eIF-2α, and p-IRE-1α in MH-S cells treated with thalidomide, SiO_2_, and SiO_2_ plus thalidomide measured by Western blot. Data are presented as the mean ± SD, *n* = 3 per group.

**Figure 7 ijms-23-05656-f007:**
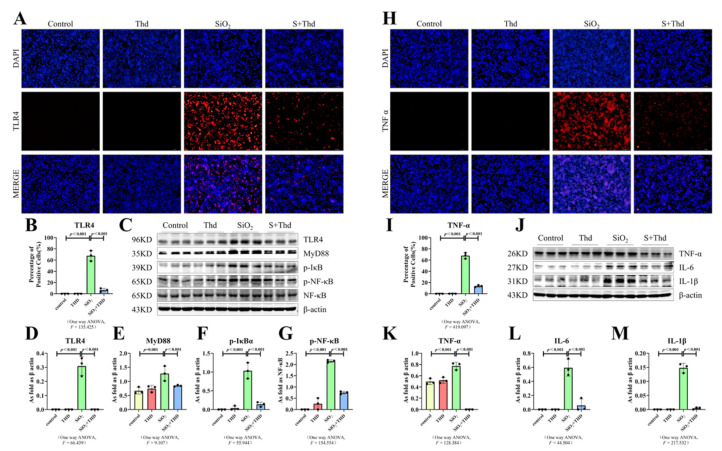
Thalidomide inhibited the TLR4-NF-κB pathway and the expression of inflammatory factors in macrophages. (**A**,**B**,**H**,**I**) Expression of TLR4 and TNF-α in MH-S cells observed by IF staining (scale bar = 100 µm). (**C**–**G**,**J**–**M**) Protein expression of TLR4, MyD88, p-IκBα, p-NF-κB TNF-α, IL-6, and IL-1β in MH-S cells treated with thalidomide, SiO_2_, and SiO_2_ plus thalidomide measured by Western blot. Data are presented as the mean ± SD, *n* = 3 per group.

**Figure 8 ijms-23-05656-f008:**
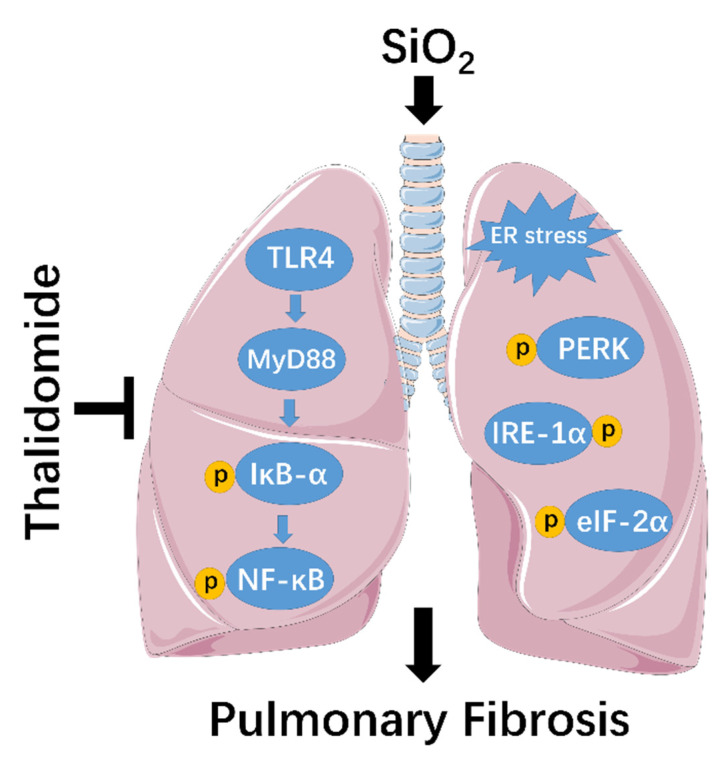
Thalidomide alleviates pulmonary fibrosis induced by silica in mice by inhibiting ER Stress and the TLR4-NF-κB pathway.

## Data Availability

Not applicable.

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
