# Peer review of "Thalidomide Alleviates Pulmonary Fibrosis Induced by Silica in Mice by Inhibiting ER Stress and the TLR4-NF-κB Pathway"

_ijms, 2022, doi:10.3390/ijms23105656_

Round 1

Reviewer 1 Report

The manuscript “Thalidomide Alleviates Pulmonary Fibrosis Induced by Silica in Mice by Inhibiting ER Stress and the TLR4-NF-κB Pathway” by Yaqian Li et al. They have; reported that Thalidomide effectively attenuates silica-induced pulmonary inflammation and fibrosis in mice and MH-S cells model. The manuscript is written in good English language with few grammatical errors. I have just a few minor concerns.

Comments:

  1. The authors should check grammatical errors very carefully before resubmission.
  2. Resolution of figure labels of 2 E-H, 3C-E, 4B-K, 5A-M, 6B-F, and 7B-M are poor and must be improved

Reviewer 2 Report

This is a well done study. The questions posed by the authors have been convincingly proven experimentally.

I have no serious comments. But there are a few wishes.
1. The quality of the figures needs to be improved. Enlarge diagrams as they are difficult to read.
2. Since the authors are talking about the mechanism, I recommend preparing an intracellular signaling scheme at the end of the article.
3. The work should be carefully re-read, because there are small errors, extra commas, etc.
